# Exposure to Green and Historic Urban Environments and Mental Well-Being: Results from EEG and Psychometric Outcome Measures

**DOI:** 10.3390/ijerph192013052

**Published:** 2022-10-11

**Authors:** Rebecca Reece, Anna Bornioli, Isabelle Bray, Chris Alford

**Affiliations:** 1Centre for Public Health and Wellbeing, University of the West of England, Bristol BS16 1QY, UK; 2Erasmus Centre for Urban, Port and Transport Economics, Erasmus University Rotterdam, Burgemeester Oudlaan 50, 3062 PA Rotterdam, The Netherlands; 3Psychological Sciences Research Group, University of the West of England, Bristol BS16 1QY, UK

**Keywords:** green environments, historic environments, well-being, EEG, virtual exposure, restorative environments

## Abstract

Previous studies have identified the benefits of exposure to green or historic environments using qualitative methods and psychometric measures, but studies using a combination of measures are lacking. This study builds on current literature by focusing specifically on green and historic urban environments and using both psychological and physiological measures to investigate the impact of virtual exposure on well-being. Results from the psychological measures showed that the presence of historic elements was associated with a significantly stronger recuperation of hedonic tone (*p* = 0.01) and reduction in stress (*p* = 0.04). However, the presence of greenness had no significant effect on hedonic tone or stress. In contrast, physiological measures (EEG) showed significantly lower levels of alpha activity (*p* < 0.001) in occipital regions of the brain when participants viewed green environments, reflecting increased engagement and visual attention. In conclusion, this study has added to the literature by showing the impact that historic environments can have on well-being, as well as highlighting a lack of concordance between psychological and physiological measures. This supports the use of a combination of subjective and direct objective measures in future research in this field.

## 1. Introduction

It is vital to understand how exposure to different environments can impact mental well-being. Globally, citizens have poor mental health with around 5% of adults suffering from depression [1] and in the UK, 74% of people suffer from high levels of stress [2]. Despite the benefits of living in urban settings, including increased work and social opportunities and access to services, research has found that living in an urbanised area can have a negative impact on mental health [3,4,5], and can be a risk factor for mental health conditions, including major depression and schizophrenia [6]. This is a public health concern, given that 55% of the global population lives in urban settings [7].

On the other hand, the well-being-benefits of spending time in nature are well-known [8,9,10]. Importantly, according to the literature, exposure to urban green infrastructure can be beneficial for the health and well-being of urban populations [11,12,13]. These benefits include stress reduction [14], positive self-reported well-being [15], improved cognitive function [16] and more. Therefore, there is growing consensus on the notion that incorporating natural elements in cities can be a successful strategy to improve mental well-being at the population level.

Further, exposure to certain environments through virtual methods has also been found to be beneficial for well-being [17,18,19] and can lead to restorative benefits [20,21,22]. Viewing environments virtually can be beneficial for populations with limited mobility and access to real environments (e.g., older people, prisoners) or in clinical settings, and it is also an important tool in researching the benefits of exposure to different environments [23,24].

From a theoretical point of view, benefits from natural environments have been explained by several psychological theories [25,26]. Attention Restoration Theory (ART) explains exposure to natural environments as being a restorative experience, allowing for the recovery of directed attention and mental restoration [25]. Natural environments allow for attention to be captured in a ‘softly’ fascinating way but do not deplete attentional resources [27]. Further, Stress Recovery Theory (SRT) states that exposure to natural environments can allow for recovery from stressful situations [26].

Some studies have shown that built settings without the presence of traffic, and with no nature, can also be restorative and support mental well-being. These include squares [28], streets [29,30,31], open built spaces [32], and museums [33,34]. From a theoretical perspective, these benefits can also be explained by ART. According to ART, any environment which presents the restorative properties of being away, fascination, compatibility and extent can offer restorative benefits [25]. Historic environments have emerged as a specific typology of built places which have a high restorative potential [35,36]. Numerous studies have suggested that historic places, including houses of worship [37,38], museums [33,34] and historic urban spaces [32] can support mental well-being and restoration.

According to the evidence, these potential benefits from exposure to historic environments are due to two types of characteristics. First, the objective characteristics of historic buildings, including the geometry, can contribute to restoration [39], with nature known to encompass a rich fractal content (bottom-up characteristics include objects’ shapes, sizes, and luminance) which can lead to ‘effortless looking’, as explained by ART [25] and reduce stress [40]. This fractal fluency, and ‘effortless looking’ can occur in built and historic environments, as well as natural environments [40], and it is the fractal elements which are used to design architecture in the first place [39]. Second, subjective, top-down features might also explain the mental well-being benefits of exposure to historic settings. Perceptions, associations and memories related to cultural identity and spirituality triggered by historic buildings can support restorative experiences and positive affective outcomes [38,41,42]. Exposure to environments which present both natural and historic elements might be especially beneficial for mental well-being, as green and historic environments might be, at the same time, quiet and interesting. In line with this idea, previous research found mental well-being benefits in historic cemeteries [43], a rural monastery [38], and a cathedral courtyard [29].

Previous studies attesting to the well-being benefits of exposure to historic settings employ qualitative methods [43,44], psychometrics [29], and surveys and perceptual scales [32]. However, there is a lack of studies examining physiological measures and/or a combination of psychometric and physiological measurements. Physiological outcomes have the advantage of providing an objective and precise measure of participants’ responses, which are not subject to participant bias in expression. Such measures include not only traditional physiological variables, such as skin conductance and heart rate but also electroencephalogram (EEG). The latter allows for the recording of brain activity and can detect stress and arousal [45,46]. In addition, the combination of physiological and psychometric measures can provide a more complete picture and allow for the assessment of the extent to which they are aligned, and also if questionnaires provide concordant measures of participants’ well-being. In a previous study, we used a combination of physiological and psychological measures to assess mental well-being when participants were exposed to natural green and blue environments, and a built historic environment [17]. In this experiment, we found that all three environments were associated with well-being benefits, although there was no clear concordance between the physiological and psychological measures in terms of which was the most effective at improving well-being. This study showed the importance and relevance of looking at both subjective and objective outcomes, as they can provide different results which can influence recommendations made in the field of research.

In contrast, the current study assessed the impact of exposure to four urban settings with varying levels of green and historic elements, all protected from motor traffic, on mental well-being, also combining psychometric and physiological measurements. The first aim was to assess whether exposure to urban environments with green and historic elements and their combination, supported mental well-being. The second aim related to the agreement between the outcome measures used. The research questions posed for this study were:

RQ1: Does exposure to urban environments with green and historic elements, and their combination, benefit mental well-being? 

RQ2: Do psychometric and physiological measurements agree when assessing the well-being benefits of exposure to different urban environments?

## 2. Materials and Methods

### 2.1. Participants

35 healthy undergraduate psychology students with normal or corrected-to-normal vision were recruited from within a UK university via the psychology participant pool between October 2019 and January 2020. Most participants were female (Table 1). Participants who signed up for the study were screened for exclusion criteria. Due to the use of electrodes on the skin for measuring EEG, participants were excluded if they suffered from a skin allergy or hypersensitive skin, had experienced an epileptic seizure within the past 12-months, had current or previous hypertension, anxiety, psychiatric, neurological disorders, or current illness (e.g., Influenza), and/or were taking prescribed medication for brain or psychiatric conditions (e.g., anxiety, depression, or epilepsy).

Five participants were excluded from the EEG analysis because of failed EEG recordings, leaving a total sample of 30 datasets for EEG analysis. All participants enrolled in the study voluntarily and gave their informed consent before they participated in the study. Participants received course credits in exchange for their participation.

### 2.2. Stimuli

Based on the first aim of the study to assess the mental well-being potential of exposure to urban green and historic settings, a 2 × 2 factorial design was chosen, with the factors being greenness (green, non-green) and architectural style (historic, modern). The four conditions, green modern, green historic, non-green modern, and non-green historic, are described in Table 2. This design allowed us to estimate the effect of each characteristic (greenness and architectural style) on mental well-being variables. At the same time, the design also allowed us to test for an interaction between green and historic environments.

Four videos depicting urban traffic-free scenes, one per condition, were created for the experiment (Figure 1). Videos were filmed on several afternoons in July 2019. Weather conditions were dry and mostly sunny across all the videos. Each video lasted 4 min, and each included several 30 s sequences of environments in Bristol, UK. The videos in the green conditions (green modern and green historic) depicted built settings (rather than parks) with green elements such as trees, bushes and grass. The videos in the non-green conditions (non-green modern and non-green historic) depicted built settings of modern styles including glass and metal materials and smooth concrete surfaces. The videos in the historic conditions depicted built historic settings, including 18th-century structures, cobbled-stone pavements, and religious and historic elements (lamp posts, statues). Videos showed a low to a moderate number of people (e.g., 20 pedestrians per minute). These were projected on a 20” flat-screen computer monitor. Each video was preceded by a short video (2 min) of a motorway road with heavy traffic acting as a stressor (Figure 2). The order of the experimental videos was randomized.

### 2.3. Design

A within-subject cross-over 2 × 2 factorial design was employed, where participants served as their own control. Figure 3 provides an overview of the experimental procedure. Participants began by watching a 4-min video of the stressor, followed by baseline completion of the affect scale (Section 0). This was followed by four sections—one per condition—in which participants watched 4 min of the video condition and completed the affect scale to measure self-reported psychological outcomes. Each section terminated with a shorter stressor video (2 min) that aimed to bring participants back to a negative mood, except for Section 4 which concluded with a final questionnaire (the affect scale). Participants were randomly assigned to the order of viewing the sections, but all participants watched all videos. Physiological measures (EEG) were recorded continuously throughout the experiment. The experiment was piloted with two participants, which confirmed that no changes were needed to the length of the videos or the general procedure.

### 2.4. Measures

#### 2.4.1. Subjective Measure

Mood was subjectively measured using the shortened version of the University of Wales Institute of Technology Mood Adjective Check List (UWIST MACL) [47]. This scale includes four items (relaxed, nervous, happy, sad). All items are measured on a 4-point scale. A score for hedonic tone and stress is produced from this measure, ranging from a minimum of 4 to a maximum of 8. Following Matthews et al., [47], scores for happy and reversed scores for sad were combined to calculate a score for hedonic tone. Scores for nervous and reversed scores for relaxed were combined to calculate a score for stress. The scale was completed on Qualtrics (Qualtrics, Provo, UT, USA, 2022) after viewing each video.

#### 2.4.2. Objective Measure

EEG was measured continuously during the experiment as an objective measure of brain activity and relaxation. EEG is a reliable measure for detecting changes in brain activity during relaxing and stressful conditions [48]. Measurements were recorded using a non-invasive cap with 32 electrodes, and electrode gel which was used to ensure low impedances. The electrode channels were connected to a QuickAmp amplifier. BrainVision Recorder 2 (BrainVision Recorder, Vers. 1.23.0001, Brain Products GmbH, Gilching, Germany) was used to record the data. A sampling rate of 1000 Hz and a notch filter of 50 Hz was applied. Markers on the recording identified when traffic and environment stimuli started and finished.

BrainVision Analyzer 2 (BrainVision Analyzer, Version 2.2.0, Brain Products GmbH, Gilching, Germany) was used to analyse the data. The EEG dataset was filtered with a low cutoff at 0.1 Hz and a high cutoff at 100 Hz. For electrodes with high impedances, pooling was conducted to create a new averaged channel. The data were then segmented into sections for traffic and environment. These sections were further segmented into 10 s sections. After segmentation, the data was examined for physical artefacts (e.g., muscle movement) using a semi-automatic inspection. Segments were removed if too many artefacts were present. Eye movements (blinks) were also removed using ocular correction with independent component analysis. Next, the data were re-referenced to the common average reference, and the sampling rate was changed to 512 Hz. Segmentation was conducted again to divide the data into 2 s segments. The data was transformed from the time domain to the frequency domain using the Fast Fourier Transform (FFT).

Spectral power (μV^2^) was exported for theta (4–7 Hz), alpha (8–12 Hz), and beta (13–30 Hz) frequency bands and transferred into an Excel spreadsheet. The overall frontal activity was averaged for electrodes Fp1, Fp2, Fz, F3, F4, F7, and F8. The medial frontal activity was averaged over electrodes F3 and F4. The lateral frontal activity was averaged over electrodes F7 and F8. The left frontal activity was averaged over electrodes F3 and F7. The right frontal activity was averaged over electrodes F4 and F8. Finally, overall occipital activity was averaged over electrodes O1, Oz, and O2.

#### 2.4.3. Additional Measures

Further measures included a final questionnaire on familiarity with the environments, landscape preferences for green versus urban and modern versus historic styles, and preference for contemplation of environments, which were all measured on 5-point scales. For each environment, three words could be chosen which the participants best thought described the environment shown. This served to check that participants were perceiving the environments as intended. The choice of descriptors included: green, urban, historic, natural, old, modern, new, attractive, cultural, and built. Additionally, sociodemographic data were collected for age (in whole years), sex (female, male, other, prefer not to say), and ethnicity (17 listed options to choose from).

### 2.5. Procedure

Ethical approval was provided by the Faculty Research Ethics Committee (FREC) from The University of the West of England and was in accordance with the Declaration of Helsinki. A full risk assessment was completed and approved before the study began. Upon arrival in the lab, participants were asked to read a participant information sheet and sign an informed consent form before taking part in the study. Then, the EEG equipment was set up. Participants were seated 46 cm away from a flat-screen Hewlett–Packard computer monitor (57 × 34 cm) with a loud speaker and asked to imagine themselves in the environments that were to be shown in the videos. Participants were then shown the video sequences presented in Figure 3 while EEG measurements were recorded, completed the UWIST MACL scale after each video sequence, and then completed the final questionnaire. Finally, the participant was thanked and debriefed. The total duration of the experiment, including setup, testing and wrapping up, was around 1.5 h for each participant.

### 2.6. Data Analysis

Data were analysed using IBM SPSS Version 26 (IBM Corp., Armonk, NY, USA) and Stata 17 (Stata Corp, College Station, TX, USA). Subjective measures were analysed with linear mixed-effects models using a long-format dataset. Objective (EEG) data were analysed using repeated measures multivariate analysis of variance (MANOVA) as well as analysis of variance (ANOVA) to analyse separate variables. Due to the EEG data not being normally distributed, the log values were used in analyses. Analyses were also Bonferroni-adjusted. All analyses used two-tailed significance levels (*p* < 0.05).

## 3. Results

### 3.1. Descriptive Statistics

#### 3.1.1. Subjective Measures

Table 3 and Table 4 show the mean levels of hedonic tone and stress for the stressor (traffic) condition and the four experimental conditions. Higher scores indicate improved hedonic tone (Table 3) and greater perceived stress (Table 4). These descriptive statistics show that mean levels of hedonic tone were higher when viewing the four environment videos compared with the traffic video, and stress levels were lower.

#### 3.1.2. Additional Measures

Participants also reported their general preference for environments. Regarding preference for green or urban environments, most participants preferred green environments (*n* = 19), some participants had more of a preference for urban areas (*n* = 7), and four participants indicated no preference. When asked to indicate a preference for modern or historic environments, the majority of participants preferred historic environments (*n* = 13), some preferred modern environments (*n* = 10), and seven participants had no preference.

For each video shown, participants were required to choose a maximum of three words from a list that they would use to describe the environment shown. For the green historic environment, the three most popular descriptors chosen were ‘historic’, ‘attractive’, and ‘old’. The same three descriptors were also the most popular choices for describing the non-green historic environment. For the green modern environment, the three most popular descriptors chosen were, ‘urban’, ‘built’, and ‘green’. Finally, for the non-green modern environment, the popular descriptors were, ‘modern’, ‘built’, ‘urban’, and ‘new’ (‘urban’ and ‘new’ had the same number of responses).

At the end of the experiment, participants reported their familiarity with each of the four environments that had been shown. Responses showed that participants were least familiar with the green modern environment and most familiar with the green historic environment, as shown in Table 5.

Regarding appreciation of environments, most participants (*n* = 20) stated they would ‘sometimes’ spend time in environments just to appreciate them. Six participants stated they would ‘often’ spend time, one participant said they would ‘very often’ spend time in environments just to appreciate them, two participants were not sure, and one participant did not respond.

### 3.2. Linear Mixed-Effects Models

#### 3.2.1. Subjective Measures

General Linear Mixed Models were used to estimate the effects of exposure to green and historic urban environments and any interactions between these. In these analyses, green, historic, age and sex were treated as fixed effects, thereby controlling for age and sex. The results of the final models are presented in Table 6 (hedonic tone) and Table 7 (stress).

We found no evidence that exposure to green environments significantly improved hedonic tone or decreased stress. There is some evidence that exposure to historic environments increased hedonic tone (*p* = 0.01) and reduced stress (*p* = 0.04). There was no evidence of an interaction between green and historic, so this term was removed from the final models.

#### 3.2.2. Objective Measures

A repeated measures MANOVA showed overall effects for frequency band (*F* (2.28) = 233.47, *p* < 0.001, η^2^*p* = 0.94), environment (*F* (3.27) = 3.34, *p* = 0.034, η^2^*p* = 0.27), and brain region (*F* (5.25) = 23.09, *p* < 0.001, η^2^*p* = 0.82). Significant interaction effects were also found for frequency band × environment (*F* (6.24) = 3.78, *p* = 0.009, η^2^*p* = 0.49), frequency band × brain region (*F* (10.20) = 30.02, *p* < 0.001, η^2^*p* = 0.94), and environment × brain region (*F* (15.15) = 14.13, *p* < 0.001, η^2^*p* = 0.93).

Three separate ANOVAs were then conducted, one for each frequency band (alpha, beta, theta). A repeated measures ANOVA with a Greenhouse–Geisser correction showed that alpha power differed significantly between environments (*F* (2.45, 71.04) = 7.46, *p* = 0.001, η^2^*p* = 0.21), as well as between brain regions (*F* (1.9, 54.97) = 20.56, *p* < 0.001, η^2^*p* = 0.42). This also revealed a significant interaction for environment × brain region (*F* (6.02, 174.56) = 18.17, *p* < 0.001, η^2^*p* = 0.39). For beta power, there were no significant differences for the environment, brain regions or interaction for environment × brain region. For theta power, there was a significant difference between environments (*F* (2.59, 75.19) = 5.46, *p* = 0.003, η^2^*p* = 0.16), brain region (*F* (2.27, 65.93) = 48.92, *p* < 0.001, η^2^*p* = 0.63) and interaction environment × brain region (*F* (5.16, 149.69) = 6.22, *p* < 0.001, η^2^*p* = 0.18).

More specifically focused on brain regions, ANOVAs revealed significant differences in alpha power for occipital (*F* (3, 87) = 51.95, *p <* 0.001, η^2^*p* = 0.64), lateral frontal (*F* (3, 87) = 5.08, *p* = 0.003, η^2^*p* = 0.15), and left frontal (*F* (1.47, 71.55) = 4.75, *p* = 0.007, η^2^*p* = 0.14) regions of the brain. For occipital regions of the brain, there was lower alpha power during exposure to green modern (M = −0.02, SD = 0.25) and green historic (M = −0.12, SD = 0.25) environments (see Figure 4). For the lateral frontal and left frontal, the lowest levels of alpha power were also present during exposure to the green historic environment.

For theta power, there were significant differences for occipital (*F* (2.09, 60.69) = 5.65, *p* = 0.005, η^2^*p* = 0.16), lateral (*F* (3, 87)= 17.21, *p* < 0.001, η^2^*p*= 0.37), left (*F* (3, 87) = 7.22, *p* < 0.001, η^2^*p* = 0.20), and right frontal (*F* (3, 87) = 6.47, *p* = 0.001, η^2^*p* = 0.18) regions of the brain. For occipital regions, theta power was reduced for green modern (M = 0.133, SD = 0.22) and green historic (M = 0.130, SD = 0.21) environments, with green historic being the lowest (see Figure 4). For lateral frontal regions, green historic (M = 0.40, SD = 0.21) and non-green modern (M = 0.39, SD = 0.22) environments showed the lowest levels of theta power. For left frontal, there was reduced theta power for the non-green modern environment (M = 0.27, SD = 0.19), and for right frontal, the green historic environment had the lowest theta power (M = 0.22, SD = 0.17).

## 4. Discussion

The current study assessed the impact of exposure to green and historic urban environments on mental well-being with a combination of self-reported psychological and physiological (EEG) measures. The research questions that were posed were: (1) Does exposure to urban environments with green and historic elements, and their combination, benefit mental well-being? (2) Do psychometric and physiological measurements agree when assessing the well-being benefits of exposure to different urban environments?

Descriptive analyses of subjective measures showed that exposure to all four non-traffic urban environments contributed to the recuperation of mental well-being after exposure to the traffic environment and that their restorative potential was comparable. However, among the four environments, the presence of historic elements was associated with a significantly stronger recuperation in hedonic tone and reduction in stress. This may provide evidence that exposure to historic elements is more beneficial than exposure to modern environments. In terms of mental well-being, historic elements can support mental well-being states, and these findings help to answer our first research question. The effect of historic environments supports and extends previous research which has shown the restorative potential of historic environments [17,29,32,38,44].

Our results found greenness to not be a significant predictor of subjective well-being. This could be related to the fact that according to descriptive checks, more focus was placed on historic features even when greenness was present. The most popular descriptors chosen for the environments shown were more focused on the historic characteristics rather than the green elements, even when greenness was present (e.g., ‘historic’, ‘old’). However, when participants viewed the green environments which lacked historic features, descriptors were chosen that were more focused towards greenness. This suggests that more focus was placed on historic features even when greenness was present, which supports the subjective findings that the main effect was from historic, as per our first research question. Also, it shows that greenness is potentially valued or noticed more in the absence of historic characteristics. However, regarding preferences for environments, participants were clearer about their preference for green spaces, with most participants preferring green to urban spaces. However, they were less decisive about their preferences for historic and modern spaces. A broader range of preferences was found for historic and modern spaces. This may suggest a potential mismatch between what is valued and what is beneficial for well-being. However, the current studies used urban environments that included both green and historic elements within an urban setting (Figure 1). Where more natural environments have been assessed, we have seen preferences for green environments, as has been reported by other authors assessing more natural green environments [17,49]. Therefore, future research is required to assess the relative weightings of elements such as ‘greenness’ and historic components within a range of natural/rural, semi-rural and urban environments to assess the generalizability of the findings for promoting well-being through visual exposure or immersion.

The results from the objective measure (EEG) are not as simple to interpret, however, it can be highlighted that EEG is changing in response to viewing the different green/non-green and historic/modern environments. The results showed that the presence of greenery was significant for alpha power in the occipital region of the brain. Lower levels of alpha power were recorded when participants viewed environments with greenery compared to environments without greenery. Based on previous literature, this would indicate that participants were more engaged and visually attentive when viewing the green environments as alpha is a resting state and more abundant when the brain is relatively inactive, for example with eyes closed when not processing visual imagery in the occipital (visual) cortex [50,51,52]. This state could potentially be explained by the concept of ‘soft attention/fascination’ as described in ART [25,27], however, this is difficult to decipher and test experimentally with EEG. We might have expected an increase in frontal alpha when viewing environments that induce a relaxed state, which was not found in this study. Significant interaction effects (green x historic) were also found for theta power in the lateral frontal and left frontal regions of the brain. This increase in theta power may be reflecting a state of relaxed attentiveness, which has been shown in previous studies [53]. Furthermore, changes in EEG alpha and theta have reliably been found to be associated with relaxation and meditation [54,55,56], which may be concordant with our results. Overall, regarding our second research question, although significant changes were detected, reflecting the sensitivity of the measures within the methods employed, we have found that psychometric and physiological measures do not always align when assessing the well-being benefits of exposure to different urban environments.

This study has added to the literature by assessing both subjective and objective measures of well-being. The lack of agreement between the physiological and psychometric findings highlights the importance of using a combination of measures when researching the benefits that environments can have on well-being. It also demonstrates the complexity of EEG research and interpretation of data. The use of physiological measures within this field of research is relatively new, and more studies are needed to establish their use and clarify the interpretation of results. This study has also provided a novel angle by focusing on green and historic environments and the combination of these elements, as well as highlighting the positive impact that exposure to urban historic environments can have on well-being.

Despite interesting findings, this study has limitations. The results from this exploratory experiment should be interpreted with caution due to the imbalance in males and females, with most participants being female (83%). Further, despite being reasonable for an EEG study, the sample size could have been larger which would have allowed us to assess specific sociodemographic characteristics which may have impacted the psychometric and physiological well-being outcomes. The use of a flat-screen as an exposure method in this study may have lacked immersiveness and influenced the effectiveness of exposure to environments in this study. Future studies should continue to investigate the impact of exposure to green and historic environments but adopt technology that provides a more immersive and realistic exposure, for example, virtual reality (VR). By using VR, it would be possible to create an environment which combines both green and historic features and control and isolate features of these environments. Studies have started using VR as an exposure method and the findings lend weight to the technology as a superior exposure method compared to the use of flat-screen monitors [17]. Additionally, qualitative research which highlights the green and historic environments, or features of these environments which are preferred, could be used in future research to inform and create virtual environments. Future research should continue to investigate the impact that exposure to urban historic environments can have on well-being, following the promising results of this study.

## 5. Conclusions

In conclusion, the psychological analyses showed support for virtual exposure to historic urban environments having a beneficial effect on well-being, in particular, providing the recovery of hedonic tone and reducing stress after exposure to a stressor. The results from the physiological (EEG) measurements showed that brain activity changed as a function of viewing the different environments, but were less conclusive. More specifically, the presence of greenery, compared with no greenery, was significantly associated with decreased alpha activity in occipital regions, which could be indicative of a more visually attentive and engaged state. Due to the complexity of the physiological findings, it is recommended that these results be interpreted with caution and replicated in future research, alongside other outcome measures. Future studies can build on these findings and consider using more immersive technology for exposure, e.g., VR, as well as recruiting large sample sizes with a more representative balance of sociodemographic characteristics (e.g., males and females) which would allow for subgroup analyses. This study has added to the literature by demonstrating, the impact that green and historic environments can have on well-being and the importance of including both subjective and objective measures when researching these environments.

## Figures and Tables

**Figure 1 ijerph-19-13052-f001:**
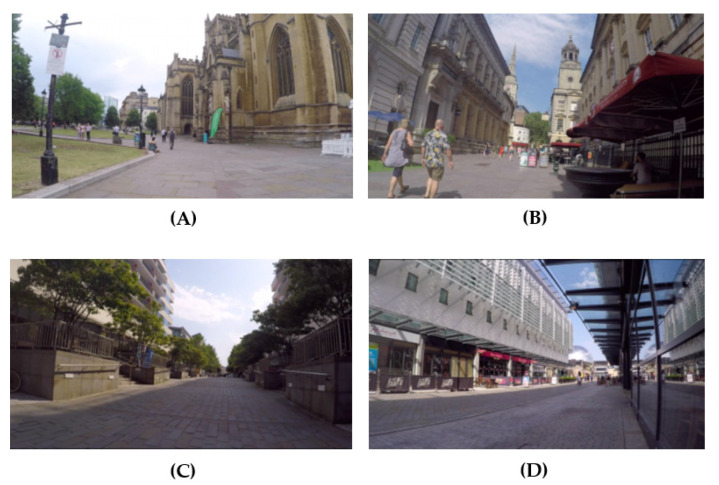
Visual stimuli for environment conditions: (**A**) green historic; (**B**) non-green historic; (**C**) green modern; (**D**) non-green modern.

**Figure 2 ijerph-19-13052-f002:**
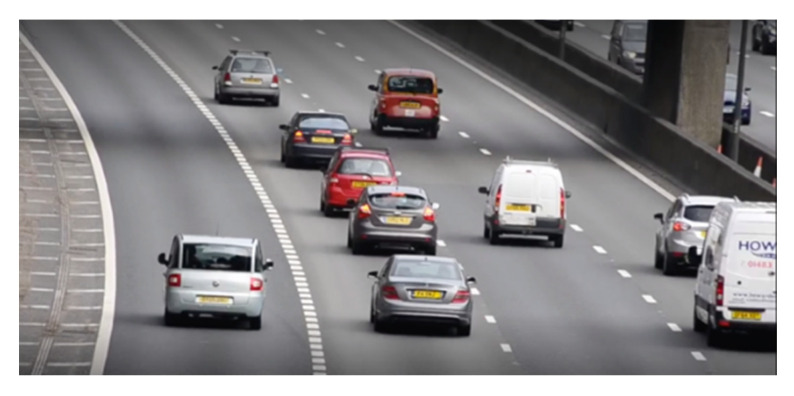
Visual stimuli for the traffic (stressor) condition.

**Figure 3 ijerph-19-13052-f003:**
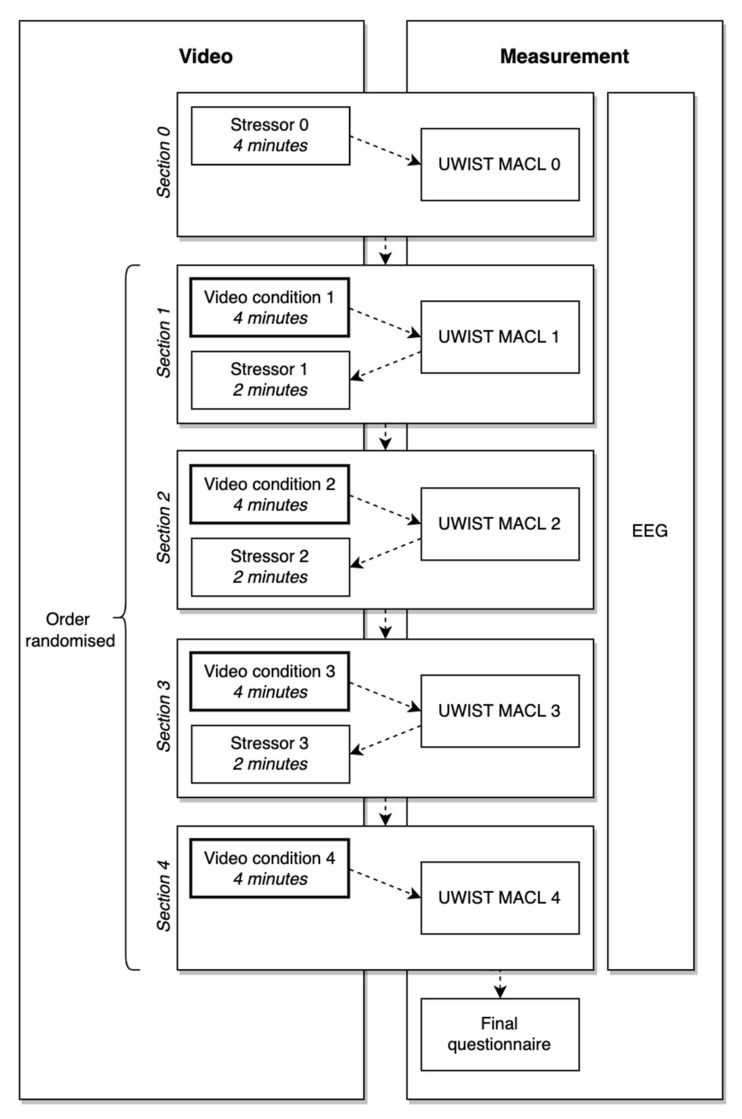
Experiment procedure.

**Figure 4 ijerph-19-13052-f004:**
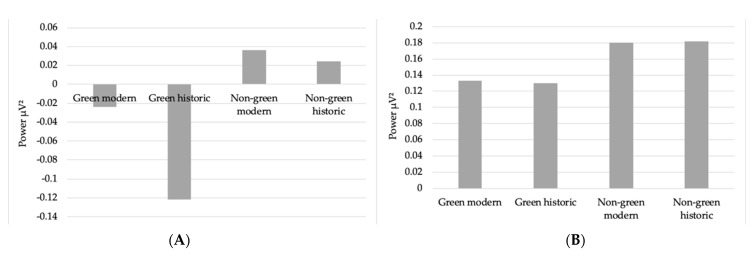
Mean occipital alpha (**A**) and theta (**B**) power across environments.

**Table 1 ijerph-19-13052-t001:** Participants’ sociodemographic characteristics.

	Median	Range	Percentage of Sample
Age (yrs)	20.0	24.0	
Male			16.7
Female			83.3
Black and Minority Ethnic			6.6

**Table 2 ijerph-19-13052-t002:** 2 × 2 Factorial design (four conditions).

	Green	Non-Green
**Historic**	Green historic (A)	Non-green historic (B)
**Modern**	Green modern (C)	Non-green modern (D)

**Table 3 ijerph-19-13052-t003:** Mean scores (SD) for hedonic tone in environment videos.

	Green	Non-Green
**Historic**	6.91 (0.93)	7.00 (0.92)
**Modern**	6.68 (0.68)	6.70 (1.31)
**Traffic**	5.88 (1.09)	

**Table 4 ijerph-19-13052-t004:** Mean scores (SD) for stress in environment videos.

	Green	Non-Green
**Historic**	2.62 (0.65)	2.62 (0.65)
**Modern**	2.82 (0.76)	2.85 (1.02)
**Traffic**	4.53 (1.33)	

**Table 5 ijerph-19-13052-t005:** Participants’ familiarity with the visual stimuli, M(SD). 5 = ‘I am very familiar’, 1 = ‘I am not familiar at all’.

	Green	Non-Green
**Historic**	3.97 (1.17)	3.91 (1.26)
**Modern**	2.32 (1.36)	3.82 (1.27)

**Table 6 ijerph-19-13052-t006:** Hedonic tone (Information Criterion 361).

Variables	Coefficient	Std. Err.	z	*p* < |z|	95% Conf. Interval	
Green	−0.09	0.12	−0.77	0.44	−0.32	0.14
Historic	0.03	0.12	2.58	0.01	0.07	0.53
Sex	−0.33	0.31	−1.07	0.28	−0.94	0.27
Age	−0.46	0.21	−2.17	0.03	−0.09	−0.00
Constant	8.40	0.88	9.56	0.00	6.68	10.13
Observations	132					
Number of groups	33					
LR test vs. linear model: chibar2(01)	30.61					
Log likelihood	−159.37					
Wald chi2(4)	12.00					
Prob > chi2	0.02					

**Table 7 ijerph-19-13052-t007:** Stress (Information Criterion 318).

Variables	Coefficient	Std. Err.	z	*p* < |z|	95% Conf. Interval	
Green	0.00	0.12	−0.00	1.00	−0.23	0.23
Historic	−0.24	0.12	−2.10	0.04	−0.47	−0.16
Sex	0.26	0.23	1.14	0.26	−0.19	0.71
Age	0.01	0.02	0.90	0.37	−0.17	0.05
Constant	2.07	0.66	0.00	0.00	0.78	3.35
Observations	132					
Number of groups	33					
LR test vs. linear model: chibar2(01)	10.89					
Log likelihood	−147.67					
Wald chi2(4)	5.92					
Prob > chi2	0.21					

## Data Availability

The data presented in this study are available on request from the corresponding author.

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
