# Peer review of "Exposure to Green and Historic Urban Environments and Mental Well-Being: Results from EEG and Psychometric Outcome Measures"

_ijerph, 2022, doi:10.3390/ijerph192013052_

Round 1

Reviewer 1 Report

The study investigated the relevance of subjective and objective indicators regarding the impact of visual exposure in green and historic urban spaces.

The research team reports that the presence of historic elements contributes to restoring mental well-being. The statistical analysis methodology is correct, and the evidence is presented.

I believe this study is an essential report on the influence of green and historical environments on well-being. I recognize this is a meaningful and suggestive study that includes developmental elements such as adopting virtual reality technology, etc., in the future.

Major Comments

Survey participants comprised 35 healthy undergraduate psychology students with normal or corrected vision.

Is there a clear rationale for limiting the study's survey participants to college students?

Is there a clear rationale for setting the sample size, which is set at 35 survey participants?

The survey participants watched a 4-minute stressor video before entering the first section and a 2-minute stressor video after the first, second, and third sections.

In this case, the interval time between sections is considered to be of critical importance.

(For example, were they conducted continuously between all sections, or was there an interval time of some minutes?)

Therefore, in describing the study's design, the authors should indicate the specific period in which the interval time settings between sections were implemented, including the reason for setting them.

This study focuses on the relationship between subjective and objective outcome measures concerning the impact of visual exposure in green and historic urban spaces.

I believe that the purposefulness of this study can be further clarified by presenting the reports of previous studies regarding the relevance of outcomes of subjective and objective indicators.

Reviewer 2 Report

The authors proposed to investigate exposure to green and historic urban environments correlated with mental well-being based on EEG results and psychometric outcome measures. The element of novelty that the authors want to bring through this research is the correlation of several measures regarding the analysis of the stability hypothesis, and not only through qualitative methods and psychometric measures. The article is well organized, and the information is presented in a coherent and logical flow. Congratulations for the work done in carrying out these interesting researches.

However, I have some small recommendations that, in my opinion, can improve the quality of the work. They will be presented in the following:

1. Figure 3. Experiment procedure does not have a very good clarity. Try to add it again in the article, with better clarity, not pixelated.

2. In section 3. Results, you used multivariate analysis of variance (MANOVA) and ANOVA for the variables selected in the model. I recommend the presentation of the methodologies used in section 2. Materials and Methods.
